# Spot Check Equivalence: an Interpretable Metric for Information Elicitation Mechanisms

## ABSTRACT

Because high-quality data is like oxygen for AI systems, effectively eliciting information from crowdsourcing workers has become a first-order problem for developing high-performance machine learning algorithms. Two prevalent paradigms, spot-checking and peer prediction, enable the design of mechanisms to evaluate and incentivize high-quality data from human labelers. So far, at least three metrics have been proposed to compare the performances of these techniques [2, 7, 29]. However, different metrics lead to divergent and even contradictory results in various contexts. In this paper, we harmonize these divergent stories, showing that two of these metrics are actually the same within certain contexts and explain the divergence of the third. Moreover, we unify these different contexts by introducing *Spot Check Equivalence*, which offers an interpretable metric for the effectiveness of a peer prediction mechanism. Finally, we present two approaches to compute spot check equivalence in various contexts, where simulation results prove the effectiveness of our proposed metric.

## CCS CONCEPTS

• **Applied computing** → *Economics*.

## KEYWORDS

Algorithmic Game Theory, Information Elicitation, Incentive for Effort, Peer Prediction

**ACM Reference Format:**
Anonymous Author(s). 2018. Spot Check Equivalence: an Interpretable Metric for Information Elicitation Mechanisms. In *Proceedings of Make sure to enter the correct conference title from your rights confirmation emai (Conference acronym 'XX)*. ACM, New York, NY, USA, 15 pages. https://doi.org/XXXXXXX.XXXXXXX

## 1 INTRODUCTION

Eliciting precise and valuable information from individuals is becoming paramount, especially with the rising demands for data labeling in the realms of AI and machine learning. Recent advancements, such as the Large Language Models (LLMs), have proven the value of high-quality human-labeled data. For example, Meta is estimated to have invested upwards of 25 million dollars collecting preference data from human labelers to align Llama 2 with human preferences [17]. This raises a pressing question: how to incentivize

human agents to provide high-quality information. E.g., without the proper incentives, human labelers for LLM allignment may not exert effort to distinguish between truthful LLM responses and merely authoritative-sounding responses (i.e. hallucinations), even when truthfulness is important for the task at hand.

Research from Amazon reveals that monetary compensation is the principal motivator for Amazon Mechanical Turk workers [3], and indeed the primary solution is to monetarily reward agents in exchange for effortful and truthful labeling. Two distinct compensation strategies, *spot-checking* [11] and *peer prediction* [19], each rates the quality of user feedback with a score. This score can then be transformed into a payment for an agent.

Spot-checking mechanisms reward agents by comparing their reports with the ground truth on a small fraction of gold standard questions. When the ground truth information is expensive or even infeasible to obtain, peer prediction mechanisms are proposed, which reward an agent based on the correlation between her reports and the reports of other agents.

To understand and compare the performance of mechanisms developed from these paradigms, there's a need for standard metrics similar to accuracy, recall, and F1 score used in supervised learning. Notice that all these metrics range from 0 to 1 where 1 is good/perfect and 0 is bad. While several studies have proposed methods for comparing these mechanisms [2, 7, 29], there remains a conspicuous gap for both a unified understanding of how these metrics relate, and, if possible, a unified interpretable metric.

To this end, we introduce the concept of *Spot Check Equivalence* (SCE), which uses a spot-checking mechanism as a benchmark to quantify the *motivational proficiency* of an arbitrary incentive mechanism. As introduced in [29], the motivational proficiency is the minimum cost of budget to induce a desired effort level in a symmetric equilibrium. Then, a SCE of 1 will indicate that a mechanism does as well as a certain spot-checking mechanism does when the spot-checking mechanism has access to the ground truth of every task. A SCE of 0 will indicate that a mechanism does as well as this same spot-checking mechanism when it has no access to the ground truth of any task (essentially, it is paying agents randomly). Note that accessing the ground truth might be costly, e.g., hiring an expert to get the ground truth might be much more expensive than hiring several non-expert crowd workers. Thus, SCE can quantify the considerable cost savings that might be achieved by employing a peer prediction mechanism over a straightforward spot-checking mechanism.

By sufficiently harshly punishing the agents for the checked low-quality reports, spot-checking mechanisms can effectively motivate effort, even when only a small fraction of the tasks are checked. However, in most real applications, the payoff should be non-negative which precludes this approach.

Gao et al. [7, 8] study a peer grading setting where agents are modeled as having a binary choice for the effort to exert: low versus high. In their setting, the goal is to minimize the fraction of

random questions that must be spot-checked to incentivize agents to exert high effort (make choosing high effort a Nash equilibrium). They find that, in their model, combining spot-checking with peer prediction does not help reduce the spot-checking ratio required to achieve the desired incentive properties, i.e. peer prediction makes things worse. However, their results rest on several assumptions, which we will discuss later in Section 6 and Appendix F.

Burrell and Schoenebeck [2] propose a metric called *Measurement Integrity* to quantify the ex-post fairness of a peer prediction mechanism. Mechanisms with high Measurement Integrity can produce payments that are strongly correlated with the quality of the agents' reports. Their motivation and definitions look beyond incentives to fairness. They do not study spot-checking mechanisms, but it is clear that the more agents are spot-checked the more accurately their scores will reflect their true quality. For example, with ground truth to all the tasks, an agent's score could exactly reflect the true quality of her responses. Moreover, they model continuous effort but do not establish a clear link between Measurement Integrity and the ability to incentivize effort.

Zhang and Schoenebeck [29] study incentivizing effort in a crowd-sourcing setting where agents can choose their effort from a continuum. Their goal is to maximize the motivational proficiency by rescaling the scores output by the incentive mechanism into the practical payments. They suggest a tournament-based payment scheme: first rank the agents based on their scores output by an incentive mechanism, and then pay a predetermined reward for each ranking. Under the tournament setting, they further propose a sufficient statistic of a mechanism's motivational proficiency, called the *Sensitivity*. Intuitively, the Sensitivity measures how responsive a score is to changes in an agent's effort. For example, a spot-checking mechanism that checks a larger fraction of tasks is more sensitive to changes in an agent's effort (intuitively, it has more chances to detect a change), and thus has a larger motivational proficiency. However, there is a lack of discussions on how to estimate Sensitivity in practice.

Here comes an apparently contradiction. Zhang and Schoenebeck [29] empirically show that when agents exert a reasonably high effort, peer prediction mechanisms have Sensitivity competitive with spot-checking mechanisms that randomly check 20% of the tasks. This seriously questions the aforementioned implication of [7, 8] as it would seem to predict that the spot-checking mechanisms are always superior to peer-prediction mechanisms.

*Our contributions.* First, we propose Spot Check Equivalence which utilizes the equivalent spot-checking ratio as an interpretable way to measure an information elicitation mechanism's performance under specified information elicitation contexts. We study the Spot Check Equivalence based on Measurement Integrity and Sensitivity, and demonstrate its effectiveness as a metric for motivational proficiency both theoretically and empirically.

Second, we unify Measurement Integrity (the metric for ex-post fairness) and Sensitivity (the metric that serves as a proxy for motivational proficiency). In particular, we prove that Spot Check Equivalence based on Measurement Integrity and Sensitivity are sometimes exactly the same. We also show why these results differ so much from Gao et al. [7, 8], and thus refute, or at least qualify, the titular statement that "Peer-prediction makes things worse."

Third, we present two approaches to compute Spot Check Equivalence, which are suitable for settings with and without ground truth data, respectively. Our method enables the comparison of the motivational proficiency of different mechanisms across various information elicitation contexts. Furthermore, our simulation results show that both approaches result in similar estimations of SCE, implying the robustness of our method.

## 2 MODEL

In this section, we will give a formal definition of the information elicitation context (Figure 1), and then formally define the Spot Check Equivalence.

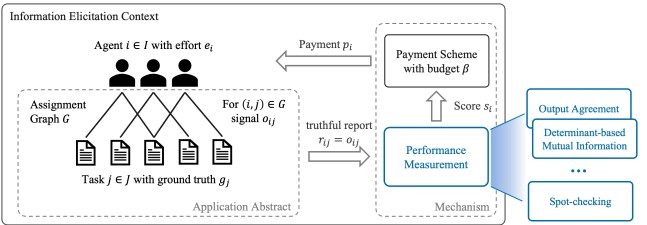

**Figure 1: Information Elicitation Context**

### 2.1 Information Elicitation Context

Formally, as shown in Figure 1, an information elicitation context (IEC) is defined as a tuple:

Information Elicitation Context $(IEC) := (Agent, App, Mech)$

where $Agent = (I, c, \mathbf{e})$ represents the agents and their properties, $App = (J, \mathcal{GT}, \omega, \Sigma, D)$ represents an information elicitation application abstraction, and $Mech = (M, P)$ represents a mechanism. We assume that the information elicitation context is common knowledge for all the agents.

*Agent.* In $Agent = (I, c, \mathbf{e})$: $I$ is the set of agents; $\mathbf{e} = [e_i]_{i \in I} \in [0, 1]^{|I|}$ represents all agents' effort levels. the cost function $c : [0, 1] \to \mathbb{R}^+ \cup \{0\}$ maps an effort level to a non-negative, increasing, and convex cost. Notice agents are homogeneous and share the same cost function.

*Application Abstraction.* $App = (J, \mathcal{GT}, \omega, \Sigma, D)$ comprises the task set $J$, the ground truth space $\mathcal{GT}$, the prior of the ground truth $\omega = \Delta_{\mathcal{GT}}$, the signal space $\Sigma$, and the data-generating process $D = (D_{assign}, D_{signal})$.

$D_{assign}$ describes how the tasks are assigned to the agents.

$$D_{assign} : \Delta_{\mathcal{G}}$$

where $\mathcal{G}$ represents the space over $G$, and $G = (I \cup J, E_G)$ represents a bipartite graph between $I$ and $J$, indicating how the tasks are assigned to the agents.

$D_{signal}$ describes how the signals are generated: the distribution of agent $i$'s signal on task $j$ conditioned on the effort $e_i \in [0, 1]$ and task $j$'s ground truth $g_j \in \mathcal{GT}$, given the edge $(i, j) \in E_G$:

$$D_{signal} : [0, 1] \times \mathcal{GT} \to \Delta\Sigma$$

*Application Instance.* With the specified *Agent* $= (I, c, \mathbf{e})$ and $App = (J, \mathcal{GT}, \omega, \Sigma, D)$, we can generate an instance representing a realized information elicitation application:

- For the given $I, J$, we sample an assignment graph $G$ according to $D_{assign}$.
- For each task $j \in J$, we independently sample its ground truth $g_j$ from the prior $\omega$.
- For each pair $(i, j) \in E_G$, we independently sample agent $i$'s signal on task $j$ from the distribution $D_{signal}(e_i, g_j)$. We then use $o_{ij} \in \Sigma$ to denote it.
- For each pair $(i, j) \in E_G$, as we assumed, the agent $i$'s report $r_{ij} = o_{ij}$.

The mechanism introduced below takes the application instance as input.

*Performance Measurement.* The performance measurement $M$ is a component of the mechanism $Mech = (M, P)$. It maps the agents' reports to their performance scores. Formally,

$$\text{(Peer Prediction)} \quad M : \Sigma^{|E_G|} \rightarrow_{random} \mathbb{R}^{|I|}$$

$$\text{(Spot-checking)} \quad M : \Sigma^{|E_G|} \times \mathcal{GT}^{|J_{\text{checked}}|} \rightarrow_{random} \mathbb{R}^{|I|}$$

Note that the spot-checking performance measurement can access the ground truth of the checked tasks $J_{\text{checked}} \subseteq J$.

We use $s_i \in \mathbb{R}$ to denote agent $i$'s score, and $\mathbf{s} = [s_i]_{i \in I}$ to denote the vector of all agents' scores.

*Payment Scheme.* The payment scheme $P$ is the other component of the mechanism. It maps the agents' performance scores to the payoffs, which are directly related to their utilities. Formally,

$$\text{(Payment Scheme)} \quad P : \mathbb{R}^{|I|} \rightarrow \left( \mathbb{R}^* \cup \{0\} \right)^{|I|}$$

We use $p_i \in \mathbb{R}^* \cup \{0\}$ to denote payoff of agent $i$, and $\beta = \sum_{i \in I} p_i$ to denote the total payment among all the agents. And we denote the vector of all the agents' payoffs as $\mathbf{p} = [p_i]_{i \in I}$.

For a better understanding of the readers, we then give two examples of the payment schemes:

**Definition 2.1 (Linear Payment Scheme).** *A linear payment scheme pays a payoff $p_i = a \cdot s_i + b$ to each agent $i$, where $a, b$ are constant parameters.*

**Definition 2.2 (Tournament Payment Scheme).** *A tournament payment scheme first ranks the agents according to their scores and then pays the i-th ranked agent $\hat{p}_i$, where $\hat{p}_1, \hat{p}_2, ..., \hat{p}_{|I|}$ are parameters and they are monotonically decreasing, i.e., $\hat{p}_i \geq \hat{p}_{i'}$ when $i \leq i'$. Without loss of generality, we assume $s_1 \geq s_2 \geq ... \geq s_{|I|}$, and thus $p_i = \hat{p}_i$.*

*Agents' Report.* We assume agents truthfully report the signals they obtain from the application abstraction conditioned on their effort levels. Further discussion will be provided in Appendix B. And we denote the agent $i$'s report on task $j$ as $r_{ij} \in \Sigma$.

*Report quality.* Given an instance, we can define the quality of an agent's report. The quality function $Q$ for one report is a deterministic loss function:

$$\text{(Quality Function)} \quad Q : \Sigma \times \mathcal{GT} \rightarrow \mathbb{R}$$

Agent $i$'s overall report quality $q_i$ is defined as the average of his reports' qualities, i.e. $q_i = \sum_{j|(i,j) \in G} Q(r_{ij}, g_j)$. We denote the vector of all the agents' qualities as $\mathbf{q} = [q_i]_{i \in I}$. Note that the report quality is not a component of an information elicitation context.

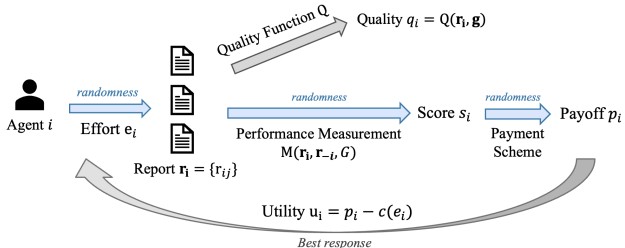

**Figure 2: An Agent's Perspective of an Information Elicitation Context**

*Equilibrium.* We then define an equilibrium in the information elicitation context.

**Definition 2.3 (Symmetric local equilibrium).** *Given IEC where all the agents exert effort $e_i = \xi$ and $\sum_{i \in I} p_i \geq |I| \cdot c(\xi)$ (Individual Rationality is satisfied), we say it is a symmetric local equilibrium if the derivative of every agent's utility is 0, i.e.*

$$\frac{\partial}{\partial e_i} u(e_i, e_{-i} = \xi)|_{e_i = \xi} = 0$$

Note that, at this equilibrium, $\frac{\partial}{\partial e_i} u(e_i, e_{-i} = \xi)|_{e_i = \xi} = 0$ is a necessary condition for $\xi$ being a Nash equilibrium. Zhang and Schoenebeck [29] show empirical evidence that local equilibrium is very likely to be Nash equilibrium in the model we mainly discuss in our paper. In the rest of our paper, our discussion will focus on this equilibrium if there's no further explanation.

## 2.2 Motivational Proficiency

The motivational proficiency of a component (a mechanism, a performance measurement, or a payment scheme) within an information elicitation context *IEC* represents its ability to incentivize effort. To quantify it, we fix all the other components of the *IEC* and quantify the expected total payment for eliciting a fixed effort level at the equilibrium (Definition 2.3), lower expected total payment implies higher motivational proficiency.

**Definition 2.4 (Motivational proficiency).** *We say the motivational proficiency of a component (Mech, M, or P) within information elicitation context IEC where all the agents exert effort level $\xi$ is higher, if and only if applying the component in IEC leads to a lower expected total payment needed to realize the symmetric local equilibrium (Definition 2.3) at effort level $\xi$.*

As we discussed in the introduction, Zhang and Schoenebeck [29] show that tournament payment schemes have higher motivational proficiency compared to linear payment schemes in certain settings, where limited liability is needed. Therefore, in the following discussion, we will focus on the motivational proficiency of performance measurements in information elicitation contexts with a tournament payment scheme. In Section 5, we quantify the

total payment in an information elicitation context with tournament payment. A simple example (Example A) in Appendix A illustrates the intuition behind many of the concepts of this paper including how the sensitivity of a score relates to the motivational proficiency of the corresponding mechanism.

## 2.3 Measure of Performance Measurements

In addition to motivational proficiency, there are other measures of performance measurements, as we discussed in the introduction, including Sensitivity [29] and Measurement Integrity [2]. We propose the general definition of a measure of performance measurement $M$.

**Definition 2.5 (Measure of Performance Measurement).** *A measure $f$ of performance measurement $M$ within information elicitation context IEC maps the IEC with $M$ to a real number, denoted as $f(IEC \leftarrow M) \in \mathbb{R}$, where the leftarrow means we apply $M$ in the information elicitation context IEC.*

We then show the two examples of measure $f$, Sensitivity [29] and Measurement Integrity [2].

*Sensitivity.* Zhang and Schoenebeck [29] propose the Sensitivity as a proxy of the motivational proficiency of a performance measurement. They show that the motivational proficiency highly depends on the Sensitivity (Definition 2.6, Lemma 2.7), which measures how an agent's performance score changes when he deviates from the equilibrium effort level.

**Definition 2.6 (Sensitivity).** *When all other agents exert effort $\xi$, we denote the mean of agent $i$'s score as $\mu_s(e_i)$, and the standard deviation as $\sigma_s(e_i)$. Then the Sensitivity of a performance measurement within an information elicitation context IEC at equilibrium effort level $\xi$ is defined as*

$$\text{Sensitivity}(IEC \leftarrow M) = \delta(\xi) = \frac{\frac{\partial}{\partial e_i}\mu_s(e_i)|_{e_i=\xi}}{\sigma_s(\xi)}$$

**Lemma 2.7.** *When all other agents exert effort $\xi$, we denote the mean of agent $i$'s score as $\mu_s(e_i)$, and the standard deviation as $\sigma_s(e_i)$. If the agent $i$' score $s_i$ follows a normal distribution $N(\mu_s(e_i), \sigma_s(e_i)^2)$, the expected total payment to elicit $\xi$ effort will (weakly) decrease in the Sensitivity $\delta(\xi)$ in a specific information elicitation context with any tournament payment scheme.*

*Measurement Integrity.* To measure the ex-post fairness of a performance measurement, Burrell and Schoenebeck [2] propose the Measurement Integrity, which is defined as the expected correlation between the quality of the agents' reports and their performance scores.

**Definition 2.8 (Measurement Integrity).** *Formally, the Measurement Integrity of a performance measurement $M$ with respect to a quality function $Q$ and a correlation function* corr, *within an information elicitation context IEC is*

$$\underset{Q,\text{corr}}{\text{MI}}(IEC \leftarrow M) = \mathbb{E}_{IEC}[\text{corr}(\mathbf{s}, \mathbf{q})]$$

## 2.4 Utilizing Spot-checking as Reference: Spot Check Equivalence

Even though we have the measures of performance measurements, there is still a need for an interpretable metric. Therefore, we utilize the checking ratio of the spot-checking performance measurement with an equal value of $f$ measure as a reference.

Firstly, we adopt the definition of spot-checking performance measurement from Gao et al. [7], and assume it can access the ground truth, thus, we call it idealized spot-checking.

**Definition 2.9 (Spot-checking performance measurement (idealized)).** *A spot-checking performance measurement is denoted as a tuple $SC := (X, S, C)$, which checks $X$ percent of all the tasks u.a.r.; then scores the agent $i$ with $S(r_{ij}, g_j)$ for each checked task $j$, and score $C \in \mathbb{R}$ for each the unchecked tasks.*

Intuitively, a higher checking ratio leads to less noise, so the high effort is easier to notice, which is beneficial for both motivational proficiency and ex-post fairness. Thus, we can use the equivalent spot-checking ratio as a metric for both motivational proficiency and ex-post fairness of a performance measurement.

Formally, for any measure of performance measurements $f$, we can define the Spot Check Equivalence as follows.

**Definition 2.10 (Spot Check Equivalence).** *For a performance measurement $M$ within an information elicitation context $IEC = (Agent, App, Mech)$, at the symmetric local equilibrium $\mathbf{e} = \xi$, we define the Spot Check Equivalence SCE of $M$, with respect to a spot-checking mechanism SC as*

$$SCE(IEC \leftarrow M) = \sup_X\{f(IEC \leftarrow M) \geq f(IEC \leftarrow SC(X, S, C))\}$$

In our following discussion, to make the spot-checking mechanism $SC(X, S, C)$ non-trivial[1], we use a quality function to score the agents for checked tasks, i.e. $S = Q$. And since we will apply a payment scheme after the performance measurement, the value of the constant score $C$ for unchecked tasks does not matter, thus, we set $C = 0$. We then use $SC(X)$ to denote $SC(X, S = Q, C = 0)$.

In the next section, we will theoretically prove the unification of Sensitivity and Measurement Integrity in certain settings, and consequently, we can show that the Spot Check Equivalence based on the Sensitivity or Measurement Integrity can be used as an interpretable metric for the motivation proficiency of an information elicitation performance measurement. We will show more empirical evidence in our agent-based model simulations (Section 5) when relaxing the theoretical assumptions.

# 3 UNIFICATION OF SENSITIVITY AND MEASUREMENT INTEGRITY

In this section, we formally prove that the Spot Check Equivalence based on Sensitivity or Measurement Integrity can be used as a proxy for Spot Check Equivalence based on motivational proficiency in certain settings. Given that computing these three measures has different requirements, the unification allows us to compute Spot Check Equivalence and consequently, measuring motivational proficiency in more scenarios.

---

[1]An example of trivial spot-checking mechanism may be $S(r_{ij}, g_j) \equiv 0$ and $C \equiv 0$.

To show this, we formally propose and prove the unification of Measurement Integrity and Sensitivity in our main theorem (Theorem 3.4). Recall that Zhang and Schoenebeck [29] have shown that, within certain settings, high Sensitivity leads to low expected total payment (Lemma 2.7) when applying a tournament payment scheme. Combining them, we can have the Spot Check Equivalence based on motivational proficiency, Sensitivity, and Measurement Integrity are equal.

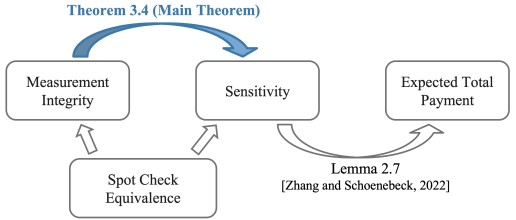

**Figure 3: Theoretical Analysis Overview**

Since the Sensitivity relies on the distribution of the performance score, we make the following assumptions about the distributions of the quantities in our model. We will further discuss the reasonability of the assumptions in Appendix C.

ASSUMPTION 3.1 (THE GAUSSIAN ASSUMPTION FOR THE QUALITY). *Given effort level $e_i$, the quality $q_i$ follows a normal distribution $N(e_i, \sigma_q(e_i)^2)$. And we further assume that $\sigma'_q(e_i) << \sigma_q(e_i)$.*

ASSUMPTION 3.2 (THE GAUSSIAN ASSUMPTION FOR THE SCORE). *Given the report quality $q_i$, the score $s_i$ follows a normal distribution $N(\mu_{s|q}(q_i), \sigma_{s|q}(q_i)^2)$.*

ASSUMPTION 3.3 (THE INDEPENDENT ASSUMPTION FOR SCORE). *When all agents have the same effort level $e_i = \xi$ and the number of agents $|I|$ goes to infinity, the agents' performance scores are independent.*

THEOREM 3.4 (MAIN THEOREM). *For a given performance measurement $M$ within an information elicitation context IEC where every agent exert effort level $\xi$, when Assumption 3.1 3.2 and 3.3 are satisfied, there exists a linear bijection between the $\mathrm{MI}_{Q,\mathrm{corr}}(IEC \leftarrow M)$ and the Sensitivity $\delta(\xi)$, where corr is the sample Pearson correlation coefficient and the number of agents goes to infinity.*

Note that our main theorem (Theorem 3.4) relies on Assumption 3.1, 3.2, and 3.3, thus, it is important to examine whether the unification of motivational proficiency, Sensitivity, and Measurement Integrity is still true with real scores calculated by various performance measurements. In the section 5, we will demonstrate some positive evidence from our agent-based model experiment, where Assumption 3.1, 3.2, and 3.3 are relaxed.

## 4  COMPUTE SPOT CHECK EQUIVALENCE

Given the unification of Sensitivity and Measurement Integrity, if we can compute the Sensitivity or Measurement Integrity of a performance measurement, we can get the Spot Check Equivalence respectively, and consequently, we can get the interpretable metric for motivational proficiency.

If the measure $f(IEC \leftarrow SC(X))$ is monotonic, we can use a binary search algorithm (Algorithm 2 in Appendix D) to compute the Spot Check Equivalence.

### 4.1  Computation with Ground Truth

We first propose a workflow to compute the Spot Check Equivalence with the ground truth of the tasks. The Spot Check Equivalence is like accuracy, recall, and F1 score in machine learning, which can only be calculated on training or testing datasets rather than real applications. Similarly, it is reasonable to create datasets to evaluate the information elicitation performance measurements, get a good sense of their motivational proficiency, and then decide which performance measurement to apply in real applications.

With the ground truth of the tasks, we can calculate the quality of the reports, and then, use the correlation between the agents' scores and qualities to estimate the measurement integrity of the performance measurement $M$ and $SC(X)$.

Intuitively, as more tasks are checked, the score $s_i$ will be more correlated to the quality $q_i$. Our agent-based model experiment also shows that the Measurement Integrity monotonically increases with the spot-checking ratio (Section 5.2). Therefore, we can apply Algorithm 2 to estimate the Spot Check Equivalence based on Measurement Integrity.

### 4.2  Computation without ground truth

Considering the current lack of information elicitation dataset, we propose another method to estimate the Spot Check Equivalence without the ground truth.

Recall the definition of Sensitivity (Definition 2.6), both the derivative of the performance score $\frac{\partial}{\partial e_i}\mu_s(e_i)|_{e_i=\xi}$ and the standard deviation $\sigma_s(\xi)$ do not require access to the ground truth. The standard deviation $\sigma_s(\xi)$ can be estimated by the standard deviation of $\{s_i\}$ given that all the agents are homogeneous. However, in real data, to estimate $\frac{\partial}{\partial e_i}\mu_s(e_i)|_{e_i=\xi}$ is tricky because the score after deviating from $\xi$ is not accessible.

We adopt the idea of bootstrap sampling: we randomly select an agent $i$, and if we know how the effort impacts the report, we can randomly manipulate his report and compute the difference between his scores before and after the manipulation. For example, if decreasing the effort brings uniform noise, we get the following algorithm.

In Section 5, we present evidence demonstrating the Algorithm 1 works in our agent-based model simulation.

## 5  EFFECTIVENESS OF THE SPOT CHECK EQUIVALENCE: AGENT-BASED MODEL EXPERIMENT

In this section, we present the results of agent-based model (ABM) experiments to evaluate the effectiveness of the *Spot Check Equivalence* based on Measurement Integrity and Sensitivity in measuring the motivational proficiency, without the assumptions we made in our theoretical proof. We then further compare different peer prediction performance measurements with spot-checking performance measurements in different information elicitation contexts.

---

**Algorithm 1:** Estimate SCE without Ground Truth

**Input:** Information Elicitation Context $IEC$, Performance
Measurement $M$, Iteration Times $T$

**Output:** Spot Check Equivalence $SCE$

**Function** *Estimate* $\Delta\mu(IEC \leftarrow M)$**:**

> $\Delta\mu = 0$
>
> **for** $t = 1$ **to** $T$ **do**
>> Choose $i \in I$ u.a.r.
>>
>> Compute score $s_i$ with $M$
>>
>> **for** $(i, j) \in \mathcal{G}$ **do**
>>> **if** $random(0, 1) > \varepsilon$ **then**
>>>> Choose $r_{ij} \in \mathcal{GT}$ u.a.r.
>>>
>>> **end**
>>
>> **end**
>>
>> Compute score $s'_i$ with $M$
>>
>> $\Delta\mu = \Delta\mu + (s'_i - s_i)/T$
>
> **end**
>
> **return** $\Delta\mu$

$SCE = \texttt{BinarySearchSCE}(M, f = \Delta\mu/\sigma_s)$

---

## 5.1 Model Setup

We first briefly introduce our agent-based model setup. The more detailed experiment setup will be discussed in Appendix E. According to the definition of the information elicitation context in Section 2, our agent-based model contains the following components:

*Agents.* We consider a population of $|I| = 50$ agents. Each agent $i$ has an effort level $e_i = \xi \in [0, 1]$ with an associated cost function $c(e_i) = e_i^2$.

*Data-generating Process (for Application Instance).* We consider a context with $|J| = 500$ tasks. Each agent is assigned to 50 tasks, while each task is assigned to 5 agents.

We adopt a generalized Dawid-Skene model from previous work [29]. The ground truth $g_j \in \mathcal{GT}$ of task $j$ is independently sampled from a discrete prior distribution $\omega$ learned from a dataset of a crowdsourcing task on Amazon Mechanical Turk [23, 29], where $\mathcal{GT}$ is a finite set including all possible ground truths.

The agent $i$ will receive a signal $o_{ij} \in \Sigma$ on task $j$ given his effort level and the task's ground truth. In our experiment, we assume that $\Sigma = \mathcal{GT}$. Then, we can define two $|\mathcal{GT}| \times |\Sigma|$ confusion matrices, $\Gamma_{work}$ and $\Gamma_{shirk}$. The $(row, col)$ entry of $\Gamma_{work}$ and $\Gamma_{shirk}$ represents the probability of getting the $row$-th signal conditioned on the $col$-th ground truth when the agent exerts effort level 1 and 0 respectively. When the agent $i$ exert $e_i$ effort, the confusion matrix is

$$\Gamma_i = e_i * \Gamma_{work} + (1 - e_i) * \Gamma_{shirk}$$

where the confusion matrix $\Gamma_{work}$ is also learned from the above dataset [23, 29], and we set $\Gamma_{shirk}$ as a matrix representing a uniform distribution.

*Performance Measurement.* We implement several peer prediction mechanisms, including Output Agreement (OA) [24–26], Peer Truth Serum (PTS) [6], Correlated Agreement (CA) [22], $f$-Mutual

Information ($f$-MI) [15, 16], and Determinant-based Mutual Information (DMI)[2] [12], which yield different Spot Check Equivalences within the above information elicitation context.

*Payment Scheme.* We now introduce the tournament Payment scheme we use in our simulation.

DEFINITION 5.1 (BORDA-COUNT PAYMENT SCHEME.). *The very intuitive way to pay an agent according to his ranking is to pay him how many agents he beats. When there's a draw, we split the payoff evenly. Formally, we have*

$$p_i = C \cdot \#beaten = C \sum_{i' \in |I|, i' \neq i} \left( \mathbb{1}[s_i > s_{i'}] + \frac{1}{2}\mathbb{1}[s_i = s_{i'}] \right)$$

*where $C$ is a constant parameter and the total payment is $C \times \binom{|I|}{2}$.*

To calculate the total payment[3] in the Borda-count scheme for a specific performance measurement $M$ for the equilibrium where every agent exerts $e_i = \xi$ effort, we set the parameter $C$ as

$$C = \frac{\frac{\partial}{\partial e_i}\mathbb{E}[\#beaten|e_i, e_{-i} = \xi]|_{e_i = \xi}}{\frac{\partial}{\partial e_i}c(e_i)|_{e_i = \xi}}$$

Note that to guarantee Individual Rationality (The agents' expected utility is non-negative), when the calculated optimal payment is less than the total cost of all the agents, we set the total payment as the total cost. We assume that if a payment scheme can incentivize effort level $\xi$ using the optimal total payment, it can also incentivize the same effort with any greater total payment.

## 5.2 Measurement-Integrity-based Spot Check Equivalence.

We examine whether the Spot Check Equivalence based on Measurement Integrity can indicate a performance measurement's motivational proficiency. Recall that the motivational proficiency of a performance measurement could be quantified by the amount of the expected total payment we need to elicit a fixed effort level in an information elicitation context. Thus, in the experiment examining the effectiveness, we mainly investigate the relationship between the Measurement Integrity and the expected total payment.

For several fixed effort levels, we enumerate the performance measurements and estimate their Measurement Integrity and the total payment needed to elicit that equilibrium with the Borda-count payment scheme by iterating the data-generating process 5000 times.

Recall that to satisfy Individual Rationality, the total payment needs to be at least the agents' cost to exert the effort. Since the minimal payment is the same for all performance measurements when the effort level is the same, we visualize that as a horizontal line in our result.

With the results in Figure 4, we can observe that:

(1) The Measurement Integrity monotonically increases with the spot-checking ratio.

---

[2]Note that even though DMI demonstrates impressive theoretical properties, it perform badly in our simulation due to the considerable noise of its score (Figure 6). Thus, we do not show it in the other figures.

[3]Note that the total payment of Borda-count is deterministic, thus, we use "total payment" in the rest of this section instead of "expected total payment".

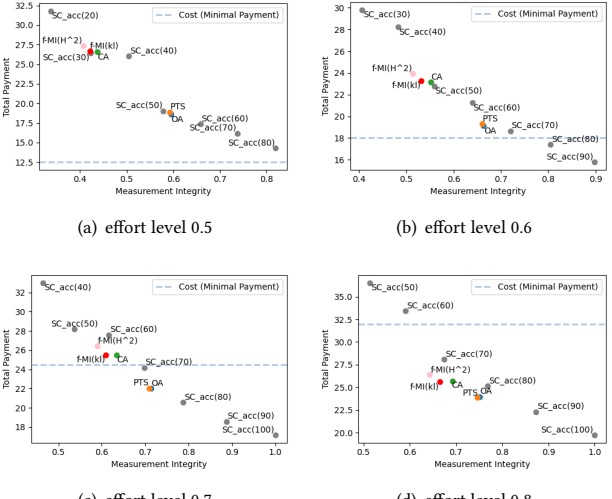

Figure 4: Measurement Integrity v.s Total Payment of Borda-count payment scheme: the $x$-axis is the Measurement Integrity and the $y$-axis is the total payment needed to elicit that equilibrium within the tournament payment scheme. The horizontal line shows the agents' cost to exert the effort level, which implies the minimal payment to satisfy the Individual Rationality.

(2) The Measurement Integrity and the total payment are significantly negatively correlated.

This implies that, at a symmetric equilibrium, if a performance measurement $M$ has Measurement Integrity equal to spot-checking $SCE\%$, then it has a similar motivational proficiency, i.e. similar total payment, to that spot-checking performance measurement within a tournament payment scheme. Therefore, a higher Spot Check Equivalence indicates higher motivational proficiency.

Note that, even if considering IR, the motivational proficiency is still monotonically increasing with the Spot Check Equivalence, however, when IR is binding, more spot-checking does not further decrease the total payment.

*5.2.1 Measurement Integrity is a computationally efficient proxy.* In addition, we find that the Measurement Integrity converges significantly faster than the total payment. The detailed result is demonstrated in Appendix E.3 and Figure 8. This suggests that even when it's possible to compute the expected total payment (e.g. with agent-based model simulation), utilizing Measurement Integrity as a proxy offers better computational efficiency.

## 5.3 Sensitivity-based Spot Check Equivalence

Similarly, we then examine the effectiveness of the Spot Check Equivalence based on Sensitivity as a metric of motivational proficiency, when there's no ground truth. We estimate the $\Delta\mu/\sigma$ (proportional to the Sensitivity) of each performance measurement according to Algorithm 1 instead of the Measurement Integrity. We then compare the $\Delta\mu/\sigma$ with the total payment estimated in the

same way as the previous subsection. We then show the results in Figure 5.

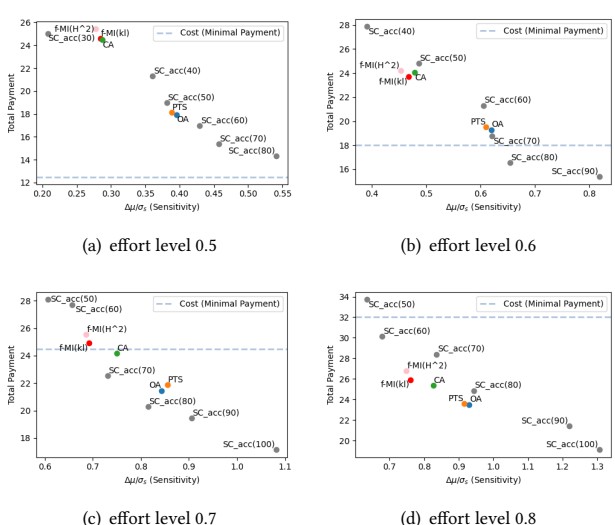

Figure 5: Sensitivity v.s Total Payment of Borda-count payment scheme: the $x$-axis is the $\Delta\mu/\sigma$ and the $y$-axis is the total payment needed to elicit that equilibrium within the tournament payment scheme. The horizontal line shows the agents' cost to exert the effort level, which implies the minimal payment to satisfy the Individual Rationality.

Similarly, With the results in Figure 5, we can observe that: (1) The Sensitivity monotonically increases with the spot-checking ratio. (2) The Sensitivity and the total payment are significantly negatively correlated. This observation implies that the Spot Check Equivalence based on Sensitivity can be used as a metric of motivational proficiency

## 5.4 Spot Check Equivalence of Peer Prediction

*5.4.1 Peer Prediction works better to elicit high effort.* We then apply the workflow in our agent-based model experiment to further compare the Spot Check Equivalence of different performance measurements in various contexts. Previous works [2, 7, 29] have shown evidence in very specific contexts. To further study the motivational proficiency of the performance measurements, we calculate the Spot Check Equivalence in various information elicitation contexts.

We then enumerate the effort levels and calculate the Spot Check Equivalence via the Measurement Integrity of each performance measurement (Figure 6).

We find that when eliciting a low-effort equilibrium the Spot-Checking Equivalence of peer prediction performance metrics is low, however, the relative motivational proficiency of peer prediction increases fast. That is because a peer prediction performance measurement scores an agent according to the correlation between his report and his peers', when every agent exerts a low effort, his peers' reports are noisy so the score is quite noisy.

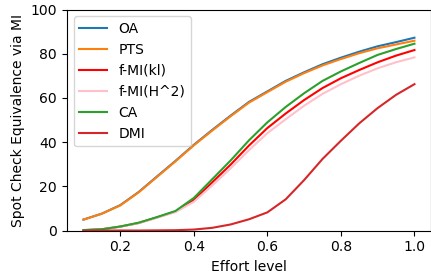

**Figure 6: Spot Check Equivalence on different effort levels**

*5.4.2 Mutual-information-based mechanisms work better when #tasks per agent increases.* As the number of tasks per agent increases, the SCE of OA and PTS remains the same, while for the mutual-information-based mechanisms (CA, $f$-MI), the Spot Check Equivalence significantly increases. As the #tasks per agent increases, the estimation of the joint distribution of two agents' reports will be more accurate, which leads to a more accurate score.

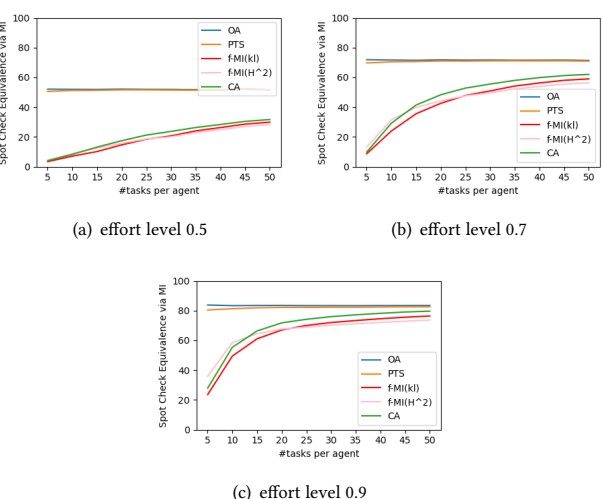

(a) effort level 0.5

(b) effort level 0.7

(c) effort level 0.9

**Figure 7: #tasks per agent vs. the Spot Check Equivalence based on Measurement Integrity: the $x$-axis is the number of tasks assigned to each agent and the $y$-axis is the Spot Check Equivalence calculated by Measurement Integrity.**

## 6 DOES PEER PREDICTION MAKE THINGS WORSE?

We now delve into the results of Gao et al. [7], especially their assertion that "peer prediction makes things worse", a claim which contradicts other studies. As we discussed in Section 1, their results rely on some restrictive assumptions.

First, they assume that payments are additive across tasks. In contrast, Zhang and Schoenebeck [29] uses tournaments, which are not linear and typically have much better motivational proficiency.

Second, they assume a fixed payment function $f$ for each task. However, as we have seen, a decrease in the spot-checking ratio can be offset by scaling payments.

Finally, the model of Gao et al. [7] assumes "cheap signals", which are signals that agents can coordinate on more easily than the signal the mechanism really desires. For example, when peer reviewing an article, it is much easier to assess quality from the author list than by meticulously reading the article to asses the quality of its argument. By reporting this "cheap signal" instead of the information desired by the mechanism agents can coordinate more with less effort and defeat certain peer-prediction mechanisms.

In response to this analysis, subsequent research has proposed peer-prediction mechanisms that aim to be robust against "cheap" signals Kong and Schoenebeck [14]. The essential idea is to ask the agents to additionally provide the cheap signals, and then pay them for coordination in addition to these signals.

A more thorough explanation is provided in Appendix F.

## 7 RELATED WORK

Besides the theoretical literature discussing peer prediction mechanisms, as highlighted in Section 5, there are empirical studies that validate these mechanisms. For instance, Radanovic et al. [21] experimentally tested their peer prediction mechanism in both peer grading and crowdsourcing scenarios to validate its theoretical properties. Similarly, Shnayder et al. [22] employed peer grading data from the edX MOOC platform to assess the performance of their proposed mechanisms. Spot checking in peer grading scenarios has already been empirically examined in works such as [27, 28]. Additionally, Goel and Faltings [9] study combining peer prediction and spot-checking, and introduce the Deep Bayesian Trust Mechanism that utilizes peer reports to reduce the need for spot-checking.

Furthermore, in forecasting contexts where agents are rewarded afterward based on the agreement between their forecasts and the outcomes, i.e. the ground truth is accessible and free, Hartline et al. [10], Li et al. [18], Neyman et al. [20] study how to optimize proper scoring rules to incentivize effort, and consequently, elicit high-quality information. These works suggest a possibility of optimizing the spot-checking mechanisms by scoring the checked tasks according to the optimal proper score rules, which indicates a future direction of our research.

## 8 CONCLUSION AND DISCUSSION

In summary, our research provides a methodology for understanding the performance, especially motivational proficiency, of information elicitation mechanisms in various contexts, the Spot Check Equivalence, and consequently offers valuable insights for the design of effective and efficient incentive mechanisms that promote the acquisition of high-quality information.

Future research might be conducted to investigate motivational proficiency in a non-monetary setting, e.g. in peer grading, we care about how to elicit agents' effort with the bounded individual payoff, since the students' grades could only be A, B, C, F, etc. Another future direction might be to study motivational proficiency in a more sophisticated model where the agents have heterogeneous cost functions.

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

## A ILLUSTRATIVE EXAMPLE

In this section, we provide a simple example that illustrates the calculation of expected total payment in an information elicitation context with a winner-take-all tournament payment scheme. We then demonstrate the Sensitivity and the Measurement Integrity in this example.

EXAMPLE A.1. *Consider a simple case where there are two agents (1 and 2) working on a large number of tasks. Each agent $i \in \{1, 2\}$ can choose to exert a non-negative effort level $e_i$, which incurs a cost $c(e_i)$. And we use a quadratic cost function, $c(e_i) = e_i^2$, which is one of the simplest functions satisfying all properties of a cost function.*

*Since the signals are independent conditioning on an agent's effort level and the spot-checking performance measurement checks each task u.a.r., according to the central limit theorem, we use normal distributions to approximate the quality and performance score: Agent $i$'s report quality is $q_i = N(e_i, 1)$, and the performance score is $s_i = N(q_i, \sigma^2)$, where $\sigma$ monotonically decreasing with the spot-checking ratio.*

*The expected utility of agent $i$ is $u_i(e_i) = \Pr[i \text{ wins the tournament}] \cdot payoff - c(e_i)$*

*To achieve the symmetric equilibrium at effort level $\xi$, we let the derivative of agent $i$'s expected utility at $e_i = \xi$ equal to 0.*

$$\frac{\partial}{\partial \xi} u_i(\xi) = 0 \Rightarrow payoff = pdf_{N(\xi, 2\sigma^2 + 2)}(\xi)$$

*where $pdf_{N(\xi, 2\sigma^2+2)}$ is the probability density function of normal distribution $N(\xi, 2\sigma^2 + 2)$.*

*In addition, we need to keep Individual Rational, i.e. the expected utility for each agent should not be negative so that the rational agents won't leave our crowdsourcing. Therefore, we have that*

$$payoff = \max\left(2c(\xi), \frac{2\xi}{pdf_{N(\xi, 2\sigma^2+2)}(\xi)}\right) = \max\left(2\xi^2, 4\sqrt{\pi \cdot (\sigma^2 + 1)} \cdot \xi\right)$$

In the above example, we can see that a higher spot-checking ratio leads to a lower total payment when IR is not binding, and consequently, a higher motivational proficiency.

*Sensitivity in Example A.1.* Recall that Zhang and Schoenebeck [29] propose the Sensitivity, which measures how sensitive the performance score is to the effort change. They show that the total payoff for eliciting effort level $\xi$ is monotonically decreasing with Sensitivity (Lemma 2.7). Here, we show the Sensitivity $\delta(\xi)$ in the above example.

$$\delta(\xi) = \frac{\frac{\partial}{\partial e_i} \mu_s(e_i)|_{e_i=\xi}}{\sigma_s(\xi)} = \frac{1}{\sqrt{\sigma^2 + 1}}$$

*Measurement Integrity in Example A.1.* In Example A.1, we can see that the motivational proficiency highly depends on how an agent's performance score correlates with his report quality. In the example, the Pearson correlation coefficient between the agent $i$'s report quality and score is as follows.

$$\rho(q_i, s_i) = \frac{\mathbb{E}[q_i \cdot s_i] - \mathbb{E}[q_i]\mathbb{E}[s_i]}{\sigma_{q_i} \cdot \sigma_{s_i}} = \frac{1}{\sqrt{\sigma^2 + 1}}$$

Note that the total payment is inversely proportional to the Pearson correlation coefficient! And the Sensitivity has the same form as the correlation in this example.

This example shows intuitions that the correlation between the agents' report qualities and scores can be a proxy for a performance measurement's motivational proficiency. And both the report quality and score are accessible in real data.

Therefore, we employ Measurement Integrity [2] as our proxy, which measures the expected correlation between the agents' report qualities and the performance scores in a specific model.

## B DISCUSSION OF THE TRUTHFUL REPORT ASSUMPTION.

In our Section 2, we assume that the agents will truthfully report their signal. This assumption is reasonable when applying a linear payment scheme given the performance measurement is truthful under certain settings (e.g. DG13 [4], CA mechanism [22], $f$-MI mechanism [15], DMI mechanism [12], etc). And Burrell and Schoenebeck [2] examine such robustness with agent-based model experiment. However, when applying a non-linear payment scheme, e.g. winner-take-all, the agents may have incentive to strategically report their signal, e.g. increasing the variance of their score to get a higher probability of being the winner. Zhang and Schoenebeck [29] propose a truthful winner-take-all payment scheme by adding noise to the agents' score which may hurt the incentive for effort. However, further study needs to be conducted to study the robustness of different performance measurements against strategic reports with other non-linear payment schemes. This gap indicates another potential future direction of our research.

## C DETAILS IN THEORETICAL ANALYSIS

### C.1 Discussion of Assumption 3.1, 3.2 and 3.3

Recall that the definition of Sensitivity relies on the distribution of the performance score. Thus, we make the Assumption 3.1, 3.2, and 3.3 to help us compute Sensitivity.

According to Central Limit Theorem, it is reasonable to make the Assumption 3.1 and 3.2 when the number of agents and the number of tasks per agent goes large. In addition, we adopt the assumption that the shape of the score distribution does not change when only deviating one agent's effort from [29].

ASSUMPTION 3.1 (THE GAUSSIAN ASSUMPTION FOR THE QUALITY). *Given effort level $e_i$, the quality $q_i$ follows a normal distribution $N(e_i, \sigma_q(e_i)^2)$. And we further assume that $\sigma_q'(e_i) << \sigma_q(e_i)$.*

ASSUMPTION 3.2 (THE GAUSSIAN ASSUMPTION FOR THE SCORE). *Given the report quality $q_i$, the score $s_i$ follows a normal distribution $N(\mu_{s|q}(q_i), \sigma_{s|q}(q_i)^2)$.*

Note $\mu_{s|q}$ is different from the $\mu_s$ in the definition of Sensitivity, in particular,

$$\mu_s(e) = \int_q \mu_{s|q}(q) \Pr[q|e] \, dq$$

We then discuss the Assumption 3.3. Note that different agents' quality $q_i$ are independent. In addition, if all agents have the same effort level $e_i = \xi$ for all $e_i$, the density of $[q_i]_{i\in I}$ will converge, as the number of agents $|I|$ goes to infinity. Formally, we have

Proposition C.1.

$$\lim_{|I| \to +\infty} \frac{\sum_{i \in I} \mathbb{1}[q_i \leq x]}{|I|} = F_{N(\xi, \sigma_q(\xi)^2)}(x) \quad \forall x$$

where $\mathbb{1}[q_i \leq x]$ is an indicator function and $F_{N(\xi, \sigma_q(\xi)^2)}(x)$ is the cumulative distribution function (CDF) of the normal distribution $N(\xi, \sigma_q(\xi)^2)$. The proof follows from the law of large numbers.

Assumption 3.3 (The independent assumption for score). *When all agents have the same effort level $e_i = \xi$ and the number of agents $|I|$ goes to infinity, the agents' performance scores are independent.*

Note that when we apply a peer-sensitive performance measurement (e.g. peer prediction), an agent's performance score relies on his report quality as well as other agents' report quality. However, Proposition C.1 shows that the distribution of all agents' report quality will converge to a fixed distribution. Therefore, it is reasonable to assume that an agent's performance score only depends on his only report quality and the fixed distribution, which implies the independence (Assumption 3.3).

## C.2 Proof of Theorem 3.4

Theorem 3.4 (Main Theorem). *For a given performance measurement $M$ within an information elicitation context $IEC$ where every agent exert effort level $\xi$, when Assumption 3.1 3.2 and 3.3 are satisfied, there exists a linear bijection between the $\mathrm{MI}_{Q,\mathrm{corr}}(IEC \leftarrow M)$ and the Sensitivity $\delta(\xi)$, where corr is the sample Pearson correlation coefficient and the number of agents goes to infinity.*

Proof. Recall the definition of Measurement Integrity, we have that

$$\mathrm{MI}_{Q,\mathrm{corr}}(IEC \leftarrow M) = \mathbb{E}_{IEC}[\mathrm{corr}(\mathbf{s}, \mathbf{q})]$$

In the above definition, we apply the *sample Pearson correlation coefficient* to corr, i.e.,

$$\mathrm{corr}(\mathbf{s}, \mathbf{q}) = r(\mathbf{s}, \mathbf{q}) = \frac{\sum_i (s_i - \bar{s})(q_i - \bar{q})}{\sqrt{\sum_i (s_i - \bar{s})^2} \sqrt{\sum_i (q_i - \bar{q})^2}}$$

where $\bar{s} = \frac{1}{|I|} \sum_i s_i$ (the average performance score); and analogously for $\bar{q}$.

Recall that we have the agents' quality independent, and the performance scores of the agents are independent when the number of agents is large (Assumption 3.3). Thus, we can regard each agent's quality and performance score pair as a sample from a joint distribution. In addition, the sample correlation coefficient $r$ is a consistent estimator of the *population Pearson correlation coefficient* $\rho$ as the sample size goes large, which is defined as

$$\mathrm{corr}(s, q) = \rho(s, q) = \frac{COV(s, q)}{\sigma_s \cdot \sigma_q}$$

where $s, q$ are random variables representing the score and quality of one agent respectively.

Therefore, we will have the following proposition.

Proposition C.2.

$$\lim_{|I| \to +\infty} \mathbb{E}[r(\mathbf{s}, \mathbf{q})] = \rho(s, q)$$

where $s, q$ are random variables representing the score and quality of one agent respectively.

We then show that there exists a bijection between $\rho(s, q)$ and $\delta(\xi)$. Recall the definition of Sensitivity

$$\delta(\xi) = \frac{\frac{\partial}{\partial e_i} \mu_s(e_i)|_{e_i = \xi}}{\sigma_s(\xi)}$$

We can find that there is $\frac{1}{\sigma_s(\xi)}$ in both the Sensitivity $\delta(\xi)$ and correlation coefficient $\rho(s, q)$, so we only need to find a bijection between $\frac{COV(s,q)}{\sigma_q}$ and $\frac{\partial}{\partial e_i} \mu_s(e_i)|_{e_i = \xi}$.

In the following discussion, since bias does not affect the correlation coefficient, we can assume $\xi = 0$ without loss of generality.

*C.2.1 From MI side.* Here, every probability is conditioned on $e = \xi$.

$$COV(s, q) = \mathbb{E}[s \cdot q] - \mathbb{E}[s] \cdot \mathbb{E}[q]$$

$$= \int_{q,s} s \cdot q \cdot \Pr[s, q] \, \mathrm{d}s \, \mathrm{d}q \qquad (\mathbb{E}[q] = \xi = 0)$$

$$= \int_{q,s} s \cdot q \cdot \Pr[s|q] \cdot Pr[q] \, \mathrm{d}s \, \mathrm{d}q$$

$$= \int_q q \cdot Pr[q] \left( \int_s s \cdot \Pr[s|q] \, \mathrm{d}s \right) \mathrm{d}q$$

$$= \int_q q \cdot Pr[q] \cdot \mu_{s|q}(q) \, \mathrm{d}q$$

*C.2.2 From Sensitivity side.*

$$\frac{\partial}{\partial e} \mu_s(e) = \frac{\partial}{\partial e} \int_q \mu_{s|q}(q) \Pr[q|e] \, \mathrm{d}q$$

$$= \int_q \mu_{s|q}(q) \frac{\partial}{\partial e} \Pr[q|e] \, \mathrm{d}q$$

Here, since $q$ follows normal distribution with mean $e$ and standard deviation $\sigma_q$, we have that

$$\frac{\partial}{\partial e} \Pr[q|e] = \frac{\partial}{\partial e} \frac{1}{\sqrt{2\pi}\sigma_q} \exp - \frac{(q-e)^2}{2\sigma_q(e)^2}$$

$$= \frac{1}{\sqrt{2\pi}\sigma_q(e)^3} \exp - \frac{(q-e)^2}{2\sigma_q(e)^2} \left( (q-e) + \frac{(q-e)^2 \sigma_q'(e)}{\sigma_q(e)} \right)$$

$$= \frac{1}{\sigma_q^2} \cdot q \cdot \Pr[q|e] \qquad \text{(Assumption 3.1)}$$

Thus, by combining the above two equations, we will get

$$\frac{\partial}{\partial e} \mu_s(e) = \int_q \mu_{s|q}(q) \cdot \frac{1}{\sigma_q^2} \cdot q \cdot \Pr[q|e] \, \mathrm{d}q$$

$$= \frac{1}{\sigma_q^2} \int_q q \cdot Pr[q|e] \cdot \mu_{s|q}(q) \, \mathrm{d}q$$

Let $e = \xi$, we will have

$$\frac{\partial}{\partial e}\mu_s(e) = \frac{1}{\sigma_q^2}\int_q q \cdot Pr[q|e = \xi] \cdot \mu_{s|q}(q)\,\mathrm{d}q$$

$$= \frac{COV(s, q)}{\sigma_q^2}$$

Therefore, we get the linear bijection between Sensitivity and correlation is

$$\delta = \mathrm{corr}(s, q)/\sigma_q.$$

□

## D COMPUTE SPOT CHECK EQUIVALENCE

In this section, we present the detailed algorithm to compute the Spot Check Equivalence when the measure $f(IEC \leftarrow SC(X))$ is monotonic with respect to $X$.

---

**Algorithm 2:** Binary Search algorithm for $SCE$

**Input:** Information Elicitation Context $IEC$, Performance
      Measurement $M$, step size $\epsilon$
**Output:** Spot Check Equivalence $SCE$
**Function** BinarySearchSCE($M, f$):
    low = 0, high = $\lfloor 100/\epsilon \rfloor$
    **while** $low \leq high$ **do**
        mid = $\left\lfloor \frac{low+high}{2} \right\rfloor$
        **if** $f(IEC \leftarrow SC(X = mid * \epsilon)) < f(IEC \leftarrow M)$ **then**
            ans = mid
            low = mid + 1
        **end**
        **else**
            high = mid − 1
        **end**
    **end**
    **return** $ans$

---

Furthermore, a linear combination of the two spot-checking ratios which has adjacent measure $f$ may be applied for a better approximation.

$$SCE = ans + \epsilon \cdot \frac{f(IEC \leftarrow M) - f(IEC \leftarrow SC(ans))}{f(IEC \leftarrow SC(ans + \epsilon)) - f(IEC \leftarrow SC(ans))}$$

## E EXPERIMENT DETAILS

### E.1 Agent-based Model Setup

In this subsection, we introduce our agent-based model setup. According to the formal definition of the information elicitation context in Section 2, our agent-based model contains the following components:

*E.1.1 Agents.* We consider a population of $|I| = 50$ agents.

*Effort and Cost.* Each agent $i$ has an effort level $e_i \in [0, 1]$ with an associated cost function $c(e_i) = e_i^2$.

*E.1.2 Data-generating Process (for Application Instance).*

*Tasks.* To show the effectiveness of Spot Check Equivalence, we consider an information elicitation application with $|J| = 500$ tasks. To compare the Spot Check Equivalence of different performance measurements in various contexts, we vary $|J|$ according to Table 1.

*Assignment graph.* The data-generating process $D$ will randomly generate a bipartite graph between $I$ and $J$: To show the effectiveness of Spot Check Equivalence, each agent is assigned to 50 tasks, while each task is assigned to 5 agents. To compare the Spot Check Equivalence of different performance measurements in various contexts, we vary the parameters according to Table 1.

| # agents | # tasks | # tasks per agent | # agents per task |
|----------|---------|-------------------|-------------------|
| 50 | 50 | 5 | 5 |
| 50 | 100 | 10 | 5 |
| 50 | $50 \times K$ | $5 \times K$ | 5 |

where $K \in \{3, 4, 5, 6, 7, 8, 9, 10\}$

**Table 1: IEC parameter setups for ABM simulations**

*Ground truth and signals.* For generating the ground truth and signals, we will apply the generalized Dawid-Skene model from previous work [5, 29], in which:

Each task $j \in J$ has a ground truth $g_j \in \mathcal{GT}$ where $\mathcal{GT}$ is a finite set including all possible ground truths.

Since $\mathcal{GT}$ is finite, we can use a vector to represent the prior distribution of the ground truth $\omega$, whose $i$th item represents the probability of $i$th possible ground truth. For convenience, we also denote that vector as $\omega$.

In our experiment, we use the $\omega$ learned from a dataset of a crowdsourcing task on Amazon Mechanical Turk [23, 29].

$$\omega = [0.19587629, 0.24054983, 0.24742268, 0.3161512]$$

The agent $i$ will receive a signal $o_{ij} \in \Sigma$ on task $j$ given his effort level and the task's ground truth. In our experiment, we assume that $\Sigma = \mathcal{GT}$. Then, we can define two $|\mathcal{GT}| \times |\Sigma|$ confusion matrices, $\Gamma_{work}$ and $\Gamma_{shirk}$. The $(row, col)$ entry of $\Gamma_{work}$ and $\Gamma_{shirk}$ represents the probability of getting the $row$-th signal conditioned on the $col$-th ground truth when the agent exerts effort level 1 and 0 respectively.

When the agent $i$ exert $e_i$ effort, the confusion matrix is

$$\Gamma_i = e_i * \Gamma_{work} + (1 - e_i) * \Gamma_{shirk}$$

where the confusion matrix $\Gamma_{work}$ is also learned from the above dataset [23, 29], and we set $\Gamma_{shirk}$ as a matrix representing a uniform distribution.

$$\Gamma_{work} = \begin{bmatrix} 0.77056673 & 0.12157221 & 0.08409506 & 0.02376600 \\ 0.09083969 & 0.73511450 & 0.12977099 & 0.04427481 \\ 0.03326256 & 0.06157113 & 0.86624204 & 0.03892427 \\ 0.06785509 & 0.16388729 & 0.09890742 & 0.66935020 \end{bmatrix}$$

*E.1.3 Performance Measurement.* We consider several performance measurements $M$ to evaluate the effectiveness of the Spot Check Equivalence. We implement several peer prediction mechanisms, which yield different Spot Check Equivalences within the above information elicitation context.

| $f(x)$ | $f$-divergence | Notation |
|---|---|---|
| $-log(x)$ | KL-divergence | KL |
| $\|x-1\|$ | Total Variation Distance | TVD |
| $(\sqrt{x}-1)^2$ | Squared Hellinger | $H^2$ |

**Table 2: $f$-Mutual Information**

*Output Agreement (OA) [24–26].* In the OA mechanism, we pair the agents who work on the same task, and compare their reports. If the reports match, both agents receive a score of 1, and if they don't, the score is 0. One agent's total score is the summation of the scores on all the tasks assigned to him.

*Peer Truth Serum (PTS) [6].* Similar to OA, we pair the agents who work on the same task, and compare their reports. Instead of scoring 1 for agreement, we score the inverse of the frequency of the agent's report, which means, intuitively, we reward more for the agreement of the uncommon reports.

*Correlated Agreement (CA) [22].* The CA mechanism is an extension of Dasgupta and Ghosh [4]'s mechanism for multi-signal information elicitation. Without loss of generality, we denote the signal (report) space as $\Sigma = [1, 2, ..., n]$. Given two agents 1 and 2, let $r_1$ and $r_2$ denote the random variables of agent 1 and 2's report on all their tasks respectively. Then, we define an $n \times n$ delta matrix $\Delta$ as

$$\Delta_{row,col} = \Pr[r_1 = row, r_2 = col] - \Pr[r_1 = row]\Pr[r_2 = col]$$

We score the agent according to the agreement on positively correlated reports, i.e. we score agent 1 and agent 2 the following value.

$$\sum_{row,col} \Delta_{row,col} \cdot \mathbb{1}[\Delta_{row,col} > 0]$$

*$f$-Mutual Information ($f$-MI) [15, 16].* The $f$-MI mechanism scores the agent with the mutual information between his and his peer's reports, the intuition is that any manipulation on the two random variables will decrease the mutual information between them. We define the $f$-Mutual Information between agent 1 and 2's report as:

$$f\text{-MI} = \sum_{row,col} \Pr[r_1 = row, r_2 = col] f\left(\frac{\Pr[r_1 = row]\Pr[r_2 = col]}{\Pr[r_1 = row, r_2 = col]}\right)$$

where $f$ is a convex function and $f(1) = 0$. For example, when we take $f(x) = -\log(x)$, then $f$-MI is the Shannon Mutual Information. Note that the $f$-Mutual Information could be regarded as the $f$-divergence ($D_f = \sum_x p(x)f\left(\frac{q(x)}{p(x)}\right)$) [1] of the joint distribution and the product of the marginal distributions of the agents' reports.

We then list all the $f$-Mutual Information we will use in our simulations in Table 2. Note that when applying a $TVD$-divergence, the $f$-MI mechanism is almost equivalent to the CA mechanism, thus, we only show the result of the CA mechanism in our simulations.

*Determinant-based Mutual Information (DMI) [12].* Kong [12] generalize the Shannon Mutual Information to the Determinant-based Mutual Information (DMI). Let an $n \times n$ matrix $\mathbf{U}_{X,Y}$ denote the joint distribution of two random variables, the DMI of these two random variables are defined as $DMI(X; Y) = |\det(\mathbf{U}_{X,Y})|$.

Then, given two agents 1 and 2, the score of the DMI mechanism is defined as the product of the DMI of the two agents' report on two disjoint partitions of all the tasks. Specifically, we divide all the common tasks of 1 and 2 into two parts, calculate their DMI in these two parts respectively, and score each agent with the product of the two DMI.

However, even though DMI demonstrates impressive theoretical property, it does not perform well in our simulation due to the considerable noise of its performance score (Figure 6).

*Spot-checking.* In addition, to calculate the Spot Check Equivalence, we also implement a spot-checking performance measurement (Definition 2.9) as a benchmark, in which we use accuracy as the quality function, i.e. $Q(r, g) = \mathbb{1}[r = g]$. We denote this spot-checking performance measurement as $SC_{ACC}$.

*E.1.4 Payment Scheme.* We now introduce the tournament Payment scheme we use in our simulation.

*Borda-count payment scheme.* The very intuitive way to pay an agent according to his ranking is to pay him how many agents he beats. When there's a draw, we split the payoff evenly. Formally, we have

$$p_i = C \cdot \#\text{beaten} = C \sum_{i' \in |I|, i' \neq i} \left(\mathbb{1}[s_i > s_{i'}] + \frac{1}{2}\mathbb{1}[s_i = s_{i'}]\right)$$

where $C$ is a constant parameter and the total payment is $C \times \binom{|I|}{2}$.

To calculate the total payment[4] in the Borda-count scheme for a specific performance measurement $M$ for the equilibrium where every agent exerts $e_i = \xi$ effort. We should set the parameter $C$ as

$$C = \frac{\frac{\partial}{\partial e_i}\mathbb{E}[\#\text{beaten}|e_i, e_{-i} = \xi]|_{e_i=\xi}}{\frac{\partial}{\partial e_i}c(e_i)|_{e_i=\xi}}$$

Note that to guarantee Individual Rationality (The agents' expected utility is non-negative), when the calculated optimal payment is less than the total cost of all the agents, we set the total payment as the total cost. We assume that if a payment scheme can incentivize effort level $\xi$ using the optimal payment, it can also incentivize the same effort when the total payment is greater than the optimal payment.

## E.2 Method

*E.2.1 Simulation for Measurement-Integrity-based Spot Check Equivalence.* To show the effectiveness of the Measurement-Integrity-based Spot Check Equivalence, we can plug different performance measurements in the information elicitation context described in the previous subsection, then calculate the Measurement Integrity (which implies the Spot Check Equivalence) and the motivational

---

[4]Note that the total payment of Borda-count is deterministic, thus, we use "total payment" in the rest of this section instead of "expected total payment".

proficiency (i.e. the total payment) respectively, and finally, observe the correlation between the Spot Check Equivalence and the motivational proficiency.

For a given information elicitation context and a target equilibrium effort level $\xi$, we propose the following workflow for one sample:

(1) Set all the agents' effort levels at $\xi$.
   (a) Generate a sample for the application abstraction according to the data-generating process $D$.
   (b) Use the performance measurement $M$ to produce all the agents' scores $\mathbf{s}$.
   (c) Use the quality function $Q$ to calculate the quality of the agents' report, and then calculate the Pearson correlation coefficient.
(2) Deviate agent 1's effort level to $e_1 = \xi - \varepsilon$
   (a) Generate another sample for the application abstraction according to the data-generating process $D$.
   (b) Use the performance measurement $M$ to produce all the agents' scores $\mathbf{s}'$.

*Calculate the Measurement Integrity.* We sample 5000 times. Use the mean of the Pearson correlation coefficient in the samples[5] as the Measurement Integrity of performance measurement $M$. We apply the same workflow to all the performance measurement in Section E.1.3, as well as spot-checking performance measurement with different checking ratios.

*Calculate the total payment.* We sample 5000 times. Use the agents' scores $\mathbf{s}$ and $\mathbf{s}'$ to estimate the $\frac{\partial}{\partial e_i}\mathbb{E}[\#\text{beaten}|e_i, e_{-i} = \xi]|_{e_i=\xi}$, which imply the total payment for the three payment schemes.

*E.2.2 Simulation for comparing SCE in various contexts.* To study how the motivational proficiency of the performance measurements changes as we vary the information elicitation context, we calculate the Spot Check Equivalence in various contexts.

We apply the same method in Section E.2.1 which can give us the Measurement Integrity of a performance measurement $M$ and the total payment needed to elicit a fixed effort level in an information elicitation context.

## E.3 Additional Results

*E.3.1 Measurement Integrity is a computationally efficient proxy.* Figure 8 illustrates the variation in both the Measurement Integrity and total payment as the number of iterations goes up.

In the previous results of Figure 4 (b), we find that when eliciting effort level of $\xi = 0.6$, the $f$-MI(kl), $f$-MI($H^2$) and CA performance measurements have a little less motivational proficiency comparable to 50% spot-checking. Meanwhile, the OA and PTS performance measurements are better than 60% spot-checking.

Figure 8 demonstrates that achieving the same outcome requires significantly fewer iterations for the calculation of Measurement Integrity compared to the total payment.

---

[5]It is a consistent estimator of the Measurement Integrity.

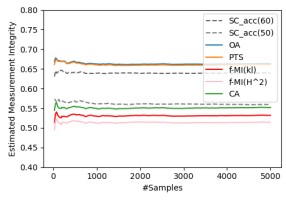 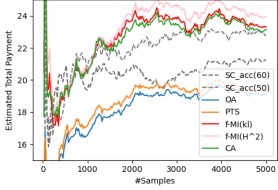

(a) Measurement Integrity  (b) Total payment of Borda-count

**Figure 8: Convergence speed of the Measurement Integrity and the total payment: the $x$-axis is the number of the samples, and the $y$-axis is the estimated Measurement Integrity and the estimated total payment of the Borda-count payment scheme at effort level $\xi = 0.6$ respectively.**

## F DOES PEER PREDICTION MAKE THINGS WORSE?

In this section, we provide a detailed discussion about why Gao et al. [7, 8] have the result that "peer prediction makes things worse" which contradicts the other literature [29] and our main results that show peer-prediction mechanisms can have non-zero Spot Check Equivalence.

We first introduce the Information Elicitation Context in their paper.

*Agent.* $Agent = (I, c, \mathbf{e})$. The agent can choose a binary effort $e \in \{0, 1\}$, where exerting high effort has a cost $c(1) = c^E$ and exerting no effort has a cost $c(0) = 0$.

*Application Abstraction.* In $App = (J, \mathcal{GT}, \omega, \Sigma, D)$, they also model signal generation as a random function:

$$D_{signal} : \{0, 1\} \times \mathcal{GT} \to \Delta\Sigma$$

however, importantly, the signals of agents are not i.i.d. across agents. When the agent exerts high effort, she will get a high-quality signal, which is drawn from a distribution conditional on the task's ground truth. When the agent exerts no effort, he will get a low-quality signal that is uncorrelated with the task's ground truth, but, crucially, the no-effort signals are perfectly correlated across no-effort agents—they all receive the same signal.

*Performance Measurement.* Firstly, they assume that the scoring function in spot-checking performance measurement (Definition 2.9) can effectively incentivize high effort, i.e.

$$\mathbb{E}\left[S\left(D_{signal}(1, g_j), g_j\right)\right] - c^E > \mathbb{E}\left[S\left(D_{signal}(0, g_j), g_j\right)\right]$$

In addition to the spot-checking performance measurement (Definition 2.9), they propose a spot-checking peer-prediction performance measurement, where for the unchecked tasks, they apply a peer-prediction performance measurement to score the agents.

*Payment Scheme.* They fix a function $f : \mathcal{GT} \times \Sigma \to \mathbb{R}^*$. The payment scheme is additive across tasks and pays agents according to $f$ for answering spot-checked tasks, and according to a peer-prediction mechanism for tasks that are not spot-checked.

*Agent reports.* They allow the agents to strategically report their signals.

*Equilibrium.* They focus on two possible equilibria. In the no-effort equilibrium, each agent exerts no effort and uses the same strategy to report his signal. In the truthful equilibrium, each agent exerts high effort and truthfully reports his signal.

## F.1 Comparing spot-checking mechanism and spot-checking peer-prediction mechanism

Notice that, by assumption, with enough spot-checking, you will get the truthful equilibrium. They use the minimum spot-checking ratio that ensures the truthful strategy profile is an equilibrium as a measure of the mechanisms' performance. Formally, they use $p_{Pareto}$ to denote the minimum spot-checking ratio where the truthful equilibrium Pareto dominates the no-effort equilibrium when applying the hybrid peer-prediction/spot-checking mechanism. They use $p_{ds}$ to denote the minimum spot-checking ratio where the truthful strategy profile is a dominant strategy in spot-checking mechanism.

Then they propose a theorem for comparing the spot-checking mechanism and the spot-checking peer-prediction mechanism:

THEOREM F.1 (SECTION 5 THEOREM 3 IN GAO ET AL. [7]). *For any spot-checking peer-prediction mechanism, if the no-effort equilibrium exists and Pareto dominates the truthful equilibrium when the cost of effort is 0 ($c^E = 0$) and no task is checked ($p = 0$), then $p_{Pareto} \geq p_{ds}$ for any $c^E \geq 0$.*

There are three differences between the models in Gao et al. [7] and Zhang and Schoenebeck [29] that account for this stark difference.

The most obvious difference is that the payments in Gao et al. [7] are restricted to be additive across tasks. This is rather similar to assuming a linear payment rule when the score is additive across tasks. However, Zhang and Schoenebeck [29] uses tournaments, which are not linear and typically have much better motivational proficiency.

Secondly, Gao et al. [7]'s assumption of a fixed payment function $f$ is extremely restrictive. As we discussed in Section 1, when the spot-checking ratio decreases, it is possible to maintain the same incentive properties by simply scaling up the payment function. Thus, if we scale up $f$, we can make $p_{ds}$ arbitrarily small, and, conversely, by scaling down $f$, we can make $p_{ds}$ arbitrarily close to 1. On the other hand, in Zhang and Schoenebeck [29] the mechanism is defined as a performance measurement and a payment scheme, and the payment scheme is optimized to work with the scoring function, rather than being artificially fixed.

Finally, Gao et al. [7]' make a key assumption on the no-effort signals being perfectly correlated. For example, in peer grading, the writing and formatting quality is a signal that can be accessed with very little effort while assessing the correctness both requires more effort and will likely lead to less agreement.

Notice that the premise of Theorem F.1 is that when the cost of effort is 0 ($c^E = 0$) and no task is checked ($p = 0$), the equilibrium where agents exert no effort Pareto dominates the truthful equilibrium.

Let's zoom in on this. First, consider the case where the cost of the high effort signal $c^E = 0$. Notice that any spot-checking mechanism that checks any positive ratio of tasks will have the high-effort profile as an equilibrium because agents will receive some positive payoff, but have 0 cost.

Next, consider a peer-prediction mechanism (e.g. a hybrid peer-prediction/spot-checking mechanism with spot-checking ratio 0). Again, the theorem is vacuous, unless the profile where no agent exerts effort Pareto dominates the high-effort equilibrium because, in the former, all agents agree and receive a maximal payoff[6], but, in the latter, they do not all agree.

Together this shows that any peer prediction mechanisms will have a Spot Check Equivalence of 0, since in this case, any spot-checking ratio that is greater than 0 will inevitably lead to the truthful equilibrium given that the effort cost is nonexistent.

Because having a Spot Check Equivalence of 0 is an assumption of the theorem it is no wonder that such peer-prediction mechanisms do not help!

However, this still potentially makes a very strong critique of peer-prediction mechanisms because in many real-world settings, there exists a cheap signal. For example, as mentioned in the introduction, when humans are labeling LLM responses, it is much easier to judge them on how authoritative-sounding the responses are than on how truthful the responses are. However, such labels may encourage hallucinations.

Indeed, Gao et al. [7] led to several papers trying to create peer-prediction mechanisms that are robust against "cheap" signals (i.e. the no-effort/low-effort signals that can bring higher agreement than the high-effort signals).

Kong and Schoenebeck [14] propose a peer prediction mechanism called Hierarchical Mutual Information Paradigm (HMIP), assuming a hierarchical information structure where high-effort (or higher expertise) agents have access to the information of low-effort (or lower expertise) agents. HMIP encourages agents to invest effort and incentivizes truthful reporting by paying the high-effort agents for correctly predicting the "cheap" signals from the low-effort agents. An empirical study [13] has also been conducted to show the evidence of the hierarchical information structure by human subject experiments.

---

[6]They are assuming here that complete agreement brings a maximum payoff.

