# OpenReview forum: "Spot Check Equivalence: an Interpretable Metric for Information Elicitation Mechanisms"
_ACM.org/TheWebConf/2024/Conference — TheWebConf24 Oral_

### Official Review · Reviewer_1yF5 · 2023-11-18

**Novelty:** 5
**Technical Quality:** 5

**Review:**

The paper develops a novel metric called Spot Check Equivalence (SCE) for measuring the performance of information gathering systems like crowdsourcing. SCE aims to assess both motivational proficiency and ex-post fairness. The authors demonstrate its effectiveness through theoretical analysis and agent-based model simulations.

Strengths
+ The proposed metric SCE provides a fresh perspective on evaluating information elicitation mechanisms. Further, SCE utilizes the spot-checking ratio as a reference point, making it interpretable.
+ Overall the paper is well written and easy to follow
+ The paper establishes that two existing metrics, sensitivity and measurement integrity, are equivalent under certain assumptions. This connection enhances the credibility of the proposed metric.

Weakenesses
- The assumptions made during SCE's developement are scattered through the text. Stating them upfront and also discussing the limitations of SCE would enhace transparency
- While theoretical analysis and simulations are valuable, empirical validation using real-world data or experiments would situate the paper’s claims better

**Questions:**

Please refer to weaknesses pointed out in teh review

**Reviewer Confidence:**

2: The reviewer is willing to defend the evaluation, but it is likely that the reviewer did not understand parts of the paper

**Scope:**

3: The work is somewhat relevant to the Web and to the track, and is of narrow interest to a sub-community

---

### Official Review · Reviewer_U4Z5 · 2023-11-20

**Novelty:** 5
**Technical Quality:** 4

**Review:**

### Summary

This work presents a methodology for understanding the effectiveness, particularly the motivational proficiency, of information elicitation mechanisms across diverse contexts, encapsulating the concept of Spot Check Equivalence. Consequently, this findings seem to contribute valuable insights for crafting incentive mechanisms that are not only effective but also efficient in fostering the acquisition of high-quality information

### Strength

1. **Introduction of Spot Check Equivalence:** The incorporation of the Spot Check Equivalence concept provides a robust framework for evaluating the comparability of different information elicitation mechanisms. This theoretical contribution offers a fresh perspective on our understanding of information incentive mechanisms.
2. **Insights into Effective and Efficient Incentive Mechanisms:** The findings of your study contribute valuable insights for the design of incentive mechanisms that are both effective and efficient. This holds practical significance for promoting the acquisition of high-quality information, with far-reaching implications for both academic and practical applications in related fields.
3. **In-Depth Exploration of the Relationship between Incentives and Performance:** Through a thorough analysis of motivational efficacy in incentive mechanisms, your study provides profound insights into our understanding of the relationship between incentives and performance.

### Weakness

1. The author mentions several metric methods in the abstract, such as [2], [7], [9], but I do not find comparative results between the proposed algorithm in this paper and these existing algorithms in the experimental section.
2. The scope of agent-based model （ABM）  seems relatively narrow, potentially impacting the generalizability of the research findings. Consider whether it is feasible to broaden the sample size or include a more diverse range of contexts to enhance the external validity of the study.
3. In the results section, the interpretation of some findings appears somewhat succinct. I suggest delving deeper into the discussion of each observed trend or relationship to ensure readers gain a more comprehensive understanding of the research outcomes.

### Mentioned references:

[2] Noah Burrell and Grant Schoenebeck. 2021. Measurement Integrity in Peer Prediction: A Peer Assessment Case Study. *arXiv preprint arXiv:2108.05521* (2021).

[7] Alice Gao, James R Wright, and Kevin Leyton-Brown. 2016. Incentivizing evaluation via limited access to ground truth: Peer-prediction makes things worse. *arXiv preprint arXiv:1606.07042* (2016).

[9] Naman Goel and Boi Faltings. 2019. Deep bayesian trust: A dominant and fair incentive mechanism for crowd. In *Proceedings of the AAAI Conference on Arti*ficial Intelligence, Vol. 33. 1996–2003.

**Questions:**

Please answer the questions to weakness 1-3.

**Ethics Review Description:**

N/A.

**Reviewer Confidence:**

2: The reviewer is willing to defend the evaluation, but it is likely that the reviewer did not understand parts of the paper

**Scope:**

3: The work is somewhat relevant to the Web and to the track, and is of narrow interest to a sub-community

---

### Official Review · Reviewer_N9WQ · 2023-11-21

**Novelty:** 5
**Technical Quality:** 5

**Review:**

## Quality
1. The background, purpose and results of the research are clearly pointed out. The equivalence of spot check is proposed, the integrity and sensitivity of measurement are unified, and two methods for calculating the equivalence of spot check are proposed, which are suitable for the setting of data with and without ground-truth. It's very detailed, and there's no obvious logical trauma.
2. The research methods and conclusions expressed are based on empirical research and theoretical analysis, and have certain scientificity and reliability. At the same time, the experimental process is also described and explained in detail, which ensures the quality of the research on the whole.

## Clarity
1. The article is clearly organized, from concept introduction, theorem derivation to model establishment, experimental analysis and result demonstration. Each section is organized in a logical order so that the reader can easily understand the author's point of view and the process of argument.
2. The concepts and terms involved in the paper are accurate, logical and easy to understand. For some professional concepts and theories, the author has also carried out appropriate explanations and explanations, so that readers can better understand the main content of the paper.
3. However, the fact that most of the content in the main text is often transferred to the appendix may not feel very friendly to beginners or readers outside the field. Details are listed in the question section.

## Originality
1. This paper proposes a new method to evaluate and motivate high-quality data, that is, by "Spot Check Equivalence" to evaluate, which has a high originality.
2. In addition, the existing evaluation indicators are compared and analyzed, which further improves their originality.
3. Finally, the author puts forward his own opinion on "peer prediction makes things worse".

## Significance
&nbsp; &nbsp; &nbsp; The problems studied in this paper are of certain importance for the development of high-performance machine learning algorithms. With the continuous development of artificial intelligence technology, the demand for high-quality data is also increasing. Therefore, how to effectively obtain high quality data from mass contractors has become an important problem. The method proposed in this paper can provide an effective tool for designers to solve this problem, which has practical significance and application value.

**Questions:**

1. Too much important content is placed in the appendix, and the text will skip too much if read only. For example, Chapter 5: how to use the confusion matrix and the detailed definition of evaluation indicators (the main text introduces f-MI, PTS and other indicators in fig4 and fig5 too briefly, which is difficult to understand if you do not read the appendix). Experimental process, algorithm 2, formula source.
2. There are many minor flaws in the writing. Please review the whole article again to confirm the use of certain words. For example, the use of single and complex numbers in line 503. For example, in lines 82 and 133, the agent's qualifier uses her, while in lines 291, 562, 563, 622, 646, 810, 1175, 1179, 1370, 1430, 1475, 1630 and 1631, the agent's qualifier changes to his.
3. Wording issues: Isn't line 93 too absolute when it says that arbitrary incentives can be measured using test equivalence? Is it appropriate to use prove in the summary (line 24), or is it better to use verify?
4. Under what circumstances might two different indicators lead to different or even contradictory results?
5. Are the scenarios considered too simple? In more complex models, how to study incentive effects when individuals have heterogeneous cost functions?
6. How to explain the use of quadratic functions in the cost function of the experiment (line 612)? Is a linear, cubic, or other function more appropriate?
7. It is mentioned in the abstract that there are three metrics (line 14), but it is not clearly pointed out which three are. After reading the whole paper, it can only be seen that what you want to unify is measurement completeness and sensitivity.
8. Is it better to give an explanation for the assumption that $r_{ij} = o_{ij}$ (line 244)?
9. You specifically state on lines 293 and 294 that report quality is not a component of the information-induced context, but you identify this component in Figure 2, and you do not say anything about Figure 2. Please give a reasonable explanation for this.
10. Some nouns are not explained. For example: What does u.a.r. stand for? Line 491 $\sigma'$ is not mentioned before and is directly used after, and the definition of IR is not mentioned in line 734.
11. It is pointed out in the appendix that hypothesis 3.1 and 3.2 are made according to the central limit theorem (line 1139), so will the scale of the experiment part be smaller (line 610 and line 615, only consider 50 agents and 500 tasks)?

**Reviewer Confidence:**

3: The reviewer is confident but not certain that the evaluation is correct

**Scope:**

3: The work is somewhat relevant to the Web and to the track, and is of narrow interest to a sub-community

---

### Official Review · Reviewer_AFvq · 2023-11-23

**Novelty:** 4
**Technical Quality:** 5

**Review:**

Summary:

The paper introduces "Spot Check Equivalence" (SCE), a metric for evaluating information elicitation mechanisms in AI and machine learning. SCE assesses how well these mechanisms perform compared to a baseline spot-checking method under different conditions. It incorporates concepts like motivational proficiency and uses metrics such as Sensitivity and Measurement Integrity. The paper demonstrates SCE's application through simulations and experiments, showing its effectiveness in different scenarios. This research offers a new framework for improving data quality in machine learning contexts.

Pros:
* The paper addresses an important problem in AI systems - effectively eliciting information from crowdsourcing workers.
* The paper presents a logical flow of ideas, systematically introducing the concept of Spot Check Equivalence (SCE) and its relevance in the context of information elicitation.
* The paper unifies different metrics and provides a comprehensive understanding of how they relate to each other.

Cons:
1. Real-world scenarios often involve agents with heterogeneous cost functions. However, this paper does not consider it.
2. The paper could benefit from more evaluation and ablation studies to validate the proposed method.

**Questions:**

1. Could you provide more insights into how SCE compares with other existing metrics in the field? Were there specific metrics that you found SCE to be particularly more effective or less effective against?
2. Can you provide more details on the implementation of the proposed approaches for computing spot check equivalence?

**Ethics Review Description:**

n.a.

**Reviewer Confidence:**

2: The reviewer is willing to defend the evaluation, but it is likely that the reviewer did not understand parts of the paper

**Scope:**

3: The work is somewhat relevant to the Web and to the track, and is of narrow interest to a sub-community

---

### Official Review · Reviewer_9KMv · 2023-11-23

**Novelty:** 4
**Technical Quality:** 5

**Review:**

This paper introduces the concept of Spot Check Equivalence, which combines a spot-checking mechanism with peer prediction as a standard to measure the motivational effectiveness of any incentive mechanism. The authors evaluate SCE using two criteria, Measurement Integrity and Sensitivity, showcasing its effectiveness as a metric for measuring motivational proficiency.

Strengths:

* S1: This paper addresses a significant challenge associated with assessing and motivating human laborers to produce high-quality data.

* S2: The authors showcased existing methods and highlighted the distinctions of their approach through experiments.

Weaknesses:

* W1: Part of the introduction could potentially be better placed within the related work section. Although referencing previous work and its limitations is valuable, reducing the depth of detail in the introduction might improve its flow.

* W2: The paper lacks a demonstration of how their approach performs differently across various tasks or contexts. While I might have overlooked it, the authors didn't explicitly exhibit in their evaluation the specific types of contexts they addressed and how their approach might excel in certain contexts more than others.

* W3:The absence of references to metrics beyond Measurement Integrity and Sensitivity, as well as the rationale behind selecting these metrics over others, has not been addressed (I added a question about this in the questions section below)

**Questions:**

* Q1: I comprehend why the authors examined Spot Check Equivalence using Measurement Integrity and Sensitivity to showcase its effectiveness as a motivational proficiency metric. However, have they explored additional metrics that might also have an impact, such as consistency, bias and fairness, or cost-effectiveness? Did the authors conduct experiments with other metrics? Should certain metrics take priority based on the task context?

* Q2: How do the authors envision their approach being applied in light of the increasing use of LLMs and the growing reliance on AI for labeling and annotating data?

* Q3: Could the authors provide additional information regarding the acquisition of ground truth through the crowdsourcing task on Amazon Mechanical Turk?

**Reviewer Confidence:**

2: The reviewer is willing to defend the evaluation, but it is likely that the reviewer did not understand parts of the paper

**Scope:**

3: The work is somewhat relevant to the Web and to the track, and is of narrow interest to a sub-community

---

### Decision · Program_Chairs · 2024-01-22

**Decision:**

Accept (Oral)

**Comment:**

Summary: Clarifies the relationships between different metrics for evaluating crowdsourcing data quality control techniques & offers a new interpretable metric.


 Strengths:
 + Addresses an important challenge of evaluating and incentivizing high-quality data from crowdsourced workers
 + A new metric (SCE) to interpret the effectiveness of peer prediction mechanisms and approaches to compute them
 + Unifies two existing metrics
 + Well-written


 Weaknesses:
 - Some aspects of writing could be improved including interpretation/discussion of results & stating assumptions/limitations clearly
 - Could demonstrate how the proposed approach varies in performance across different contexts
 - No justification why other metrics were not considered
 - No heterogeneous cost functions
 - Limited evaluations


 Recommendation: An interesting work with sound theoretical backing. Please address the issues raised by the reviewers and fulfill the promises made during the rebuttal.